# Inter-annual and decadal changes in teleconnections drive continental-scale synchronization of tree reproduction

Davide Ascoli [1], Giorgio Vacchiano [2,9], Marco Turco [3], Marco Conedera[4], Igor Drobyshev[5,6], Janet Maringer[4,7], Renzo Motta[2] & Andrew Hacket-Pain[8]

Climate teleconnections drive highly variable and synchronous seed production (masting) over large scales. Disentangling the effect of high-frequency (inter-annual variation) from low-frequency (decadal trends) components of climate oscillations will improve our understanding of masting as an ecosystem process. Using century-long observations on masting (the MASTREE database) and data on the Northern Atlantic Oscillation (NAO), we show that in the last 60 years both high-frequency summer and spring NAO, and low-frequency winter NAO components are highly correlated to continent-wide masting in European beech and Norway spruce. Relationships are weaker (non-stationary) in the early twentieth century. This finding improves our understanding on how climate variation affects large-scale synchronization of tree masting. Moreover, it supports the connection between proximate and ultimate causes of masting: indeed, large-scale features of atmospheric circulation coherently drive cues and resources for masting, as well as its evolutionary drivers, such as pollination efficiency, abundance of seed dispersers, and natural disturbance regimes.

[1] Department Agraria, University of Naples Federico II, via Università 100, 80055 Portici, Italy. [2] DISAFA, University of Turin, Largo Braccini 2, 10095 Grugliasco, TO, Italy. [3] Department Applied Physics, University of Barcelona, Av. Diagonal 647, 08028 Barcelona, Spain. [4] Swiss Federal Institute for Forest, Snow, and Landscape Research WSL, a Ramél 18, CH-6953 Cadenazzo, Switzerland. [5] Southern Swedish Forest Research Centre, Swedish University of Agricultural Sciences, P.O. Box 49 230 53 Alnarp, Sweden. [6] Institut de recherche sur les forêts, Université du Québec en Abitibi-Témiscamingue, 445 boulevard de l'Université, Rouyn-Noranda, QC J9X 5E4, Canada. [7] Institute for Landscape Planning and Ecology, University of Stuttgart, Keplerstrasse 11, 70174 Stuttgart, Germany. [8] Department of Geography and Planning, School of Environmental Sciences, University of Liverpool, Liverpool, L697ZT, UK. [9] Present address: European Commission, Joint Research Centre, D1 Bio-economy, Ispra, 21027 Varese, Italy. Correspondence and requests for materials should be addressed to D.A. (email: davide.ascoli@unina.it)

Masting, the synchronous and highly variable production of seeds by a population of plants, may periodically synchronize over large portions of a species distribution range[1–3], with major cascading effects on ecosystems functioning[4]. Large-scale masting events rely on the spatial synchronization (Moran effect) of proximate mechanisms of masting (here after "proximate causes")[5], such as those promoting resource accumulation and floral induction in the previous 1–2 years, and cross-pollination in the mast year[4]. Over longer timescales, several studies report periods with frequent large masting events that alternate with periods of rare masting and attribute this fluctuation to decadal trends in broad climate patterns[6,7]. Indeed, the interplay of proximate causes leading to masting may occur at both annual and longer time scales, e.g., by increasing the sensitivity of trees to a flowering–inducing cue during extended periods of higher resource availability due to a favorable climate trend[4,5,8,9]. Disentangling the effects of climate oscillations on masting at high (inter-annual) vs. low (decadal) frequencies would therefore result in an improved understanding of masting as an ecosystem process and its causes.

Teleconnections are broad climate patterns that produce spatially correlated weather conditions at both inter-annual and decadal time scales[10,11]. Indices describing the phase of a teleconnected climate system integrate several weather variables[12], which makes them good candidates for explaining the large-scale synchronization of ecological processes[13], including masting[1,14,15].

Several teleconnection indices were found to correlate to tree masting and its proximate causes, such as the El Niño Southern Oscillation in East Asia and Oceania[1,2,16], the North Pacific Index in North America[14], and the North Atlantic Oscillation (NAO) in Europe[7,15,17]. However, due to limitations of masting data in time and/or space, none of these studies tested for the effect of the low-frequency component of the teleconnection. Likewise, no assessment was made on whether the effects were consistent through time.

In this study, we take advantage of long-term masting observations (the MASTREE database)[18], covering most of the *Fagus sylvatica* L. (European beech) and *Picea abies* (L.) H. Karst (Norway spruce) distribution. We used MASTREE to assess the effect of inter-annual and decadal variations of the NAO on large-scale masting in both species, and the stability of such effects through time.

Beech and spruce may display synchronized reproduction over a large portion of their distribution area[3,15]. Previous studies found a relationship between NAO and beech masting in some regions of Central-Northern Europe, but uncertainties emerge regarding the timing and direction of this relationship (Table 1). Positive NAO in the winter of the year before fruit ripening ($Y_{M-1}$) favoured beech masting in Southern England[17]. Negative NAO in the summer 2 years before fruit ripening ($Y_{M-2}$), followed by a positive summer-NAO in year $Y_{M-1}$, enhanced beech masting in Southern Sweden[7], whereas positive NAO in the spring during flowering ($Y_M$) synchronized masting in Germany, France, and Luxemburg[15]. This final relationship is also reported for spruce[15]. Hence, no strong evidence of a spatially and temporally consistent influence of NAO on beech and spruce masting has emerged at the continental scale. We argue that inconsistencies in previous studies arise from the following: (a) failure to analyze the whole period during which climate affects proximate causes of masting in beech (three years from $Y_{M-2}$ to $Y_M$)[3] and spruce (2 years from $Y_{M-1}$ to $Y_M$)[19]; (b) inability to test the effects of decadal NAO components as a potential common influence on masting; and (c) disregarding that the relationship between masting and NAO may have changed through time.

In order to get a broader understanding in the relationship between masting and NAO, we address the following questions: (1) Do all seasonal NAO indices reported as relevant for masting in previous studies (Table 1) exert a significant effect on beech and spruce synchronous seed production at the European scale? (2) Do both inter-annual and decadal variations of NAO affect beech and spruce masting? (3) Are NAO–masting relationships consistent with weather patterns known to determine masting in both species? (4) Are these relationships stationary through time?

We show that in the last 60 years the inter-annual variation of NAO in summer and spring, as well as decadal trends in the winter NAO, are highly correlated to continent-wide masting in beech and spruce. This finding highlights the role of teleconnections in affecting large-scale synchronization of tree masting and provides insights on its evolutionary drivers.

## Results

**Raw seasonal NAO indices vs. masting index model**. The large-scale masting index (M_index) for beech in Central-Northern Europe (Supplementary Fig. 2, left) displayed the highest values (above 95th percentile) in years 1773, 1811, 1846, 1858, 1869, 1888, 1900, 1909, 1918, 1926, 1948, 1958, 1995, 2006, and 2011. From 1950 to 2014, the spruce M_index was significantly correlated with the beech M_index (Pearson = 0.58, $p < 0.001$, two-sided test) and several synchronized large-mast events were shared by both species (e.g., 1974, 1990, 1992, 1995, 2000, 2004, 2006, and 2011) (Fig. 1). For both species, the period used for model building showed transitions between prolonged high (i.e., in early '50 s, early '90 s of the twentieth century, and in 2010s) and prolonged low M_index (e.g., 1961 to 1986) (Fig. 1). Interestingly, this last period coincided with a reduced and non-significant correlation between the two series (Pearson = 0.35, $p = 0.12$).

**Table 1 Previous findings on the relationship between seasonal NAO indices and beech and spruce masting**

| Year before masting | $Y_{M-2}$ | $Y_{M-1}$ | $Y_{M-1}$ | $Y_M$ |
|---|---|---|---|---|
| Species | Beech | Beech | Beech | Beech–Spruce |
| NAO season | Summer | Winter | Summer | Spring |
| Correlation sign NAO season vs. masting | Negative | Positive | Positive | Positive |
| NAO phase (− or +) and weather in Europe | Summer-NAO− Weather: Cool-Wet | Winter-NAO+ Weather: Warm-Wet | Summer-NAO+ Weather: Warm-Dry | Spring-NAO+ Weather: Warm-Dry |
| Previous study | Drobyshev et al. 2014 | Piovesan & Adams 2001 | Drobyshev et al. 2014 | Fernández-M. et al. 2016 |
| Geographical area | Southern Sweden | Southern England | Southern Sweden | France, Germany, Luxemburg |
| Studied period | 1871–2006 | 1981–1995 | 1871–2006 | 2002–2010 |

Previous findings on the relationship between seasonal NAO indices (winter-NAO; summer-NAO; spring-NAO), and beech and spruce masting in different European regions. $Y_M$: year of masting; $Y_{M-1}$ and $Y_{M-2}$: 1 and 2 years before masting, respectively.

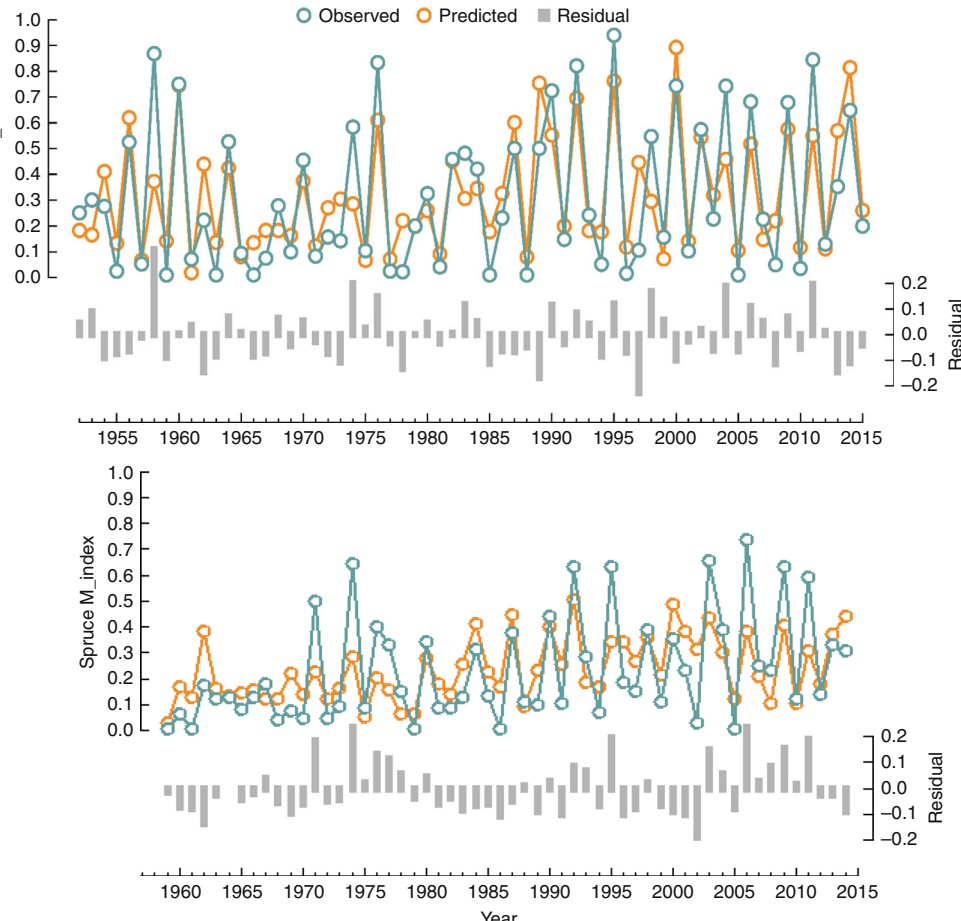

**Fig. 1** Observed and predicted values of the masting indexes. Observed (blue line) and predicted (orange line) yearly values of M_index (scaled from 0 to 1) calculated for Central and Northern Europe for beech (first row, 1950–2015) and spruce (second row, 1959–2014). Predicted values estimated according to the final model in Table 2. Gray bars are the model residuals

All the raw seasonal NAO indices tested separately and at a regional scale by previous studies (Table 1) significantly affected beech masting when analysed simultaneously. The model explained a high portion of the variability of the beech M_index between 1952 and 2015 at the sub-continental scale (pseudo-$R^2$ = 0.55, Supplementary Table 1). The summer-NAO $Y_{M-2}$ was negatively correlated with M_index, whereas the winter-NAO $Y_{M-1}$ and summer-NAO $Y_{M-1}$, and the spring-NAO $Y_M$, correlated positively. The effect of the autoregressive factor (AR1) was significant, with a negative effect (Supplementary Table 1). For spruce, only winter-NAO $Y_{M-1}$ and the spring-NAO $Y_M$ were significant and positively correlated to M_index (pseudo-$R^2$ = 0.27, Supplementary Table 1).

**Low-frequency domain of NAO and masting relationships.** Wavelet analysis showed that all seasonal NAO indices exhibited coherence with beech and spruce masting at similar low-frequency domains, although relationships displayed a different level of significance for different seasonal indices. In the second half of the twentieth century, all winter-NAO indices (National Oceanic and Atmospheric Administration (NOAA) for 1950–2015, Fig. 2a; Hurrell 1899–2015, Fig. 2b; Jones 1826–2015, Fig. 2c) showed significantly positive coherence with beech masting in the frequency domain of 7 to 16 years. A similar wavelet coherence existed between the winter-NAO using the NOAA index (1950–2015) and the spruce M_index (Supplementary Fig. 5a).

Long-term analyses (> 50 years) on beech data revealed that coherence between the beech M_index and winter-NAO varied through time. In the first half of the twentieth century, both the Hurrell (Fig. 2b) and the Jones (Fig. 2c) indices did not show any significant coherence with M_index, and only in the second half of the nineteenth century an in-phase influence of winter-NAO on the beech M_index emerged, again with a frequency domain of 7 to 16 years (Fig. 2c). Summer-NAO and M_index of both species were significantly coherent for a short period around 1980 (Supplementary Fig. 5b and Supplementary Fig. 6a and 6c), but for beech the signal in the domain of 7 to 16 years remained throughout the 20th century (albeit weakly). Finally, a significant coherence between spring-NAO and beech M_index was found using both NOAA, and Hurrell indices since 1985 at frequencies of about 8 to 16 years, but this was mostly out of the cone of influence (Supplementary Fig. 6b and 6d). For spruce, a coherence with spring-NAO remained weak around 11 years for the second half of the twentieth century (Supplementary Fig. 5c).

Despite major differences in significance and stationarity between the winter-NAO and the summer- and spring-NAO, we opted for further testing of the low-frequency component of all seasonal NAO indices. In the final regression models, low-frequency indices with a periodicity of 11 years (i.e., the midpoint between 7 and 16 years) were included for both species.

**Inter-annual and decadal NAO vs. masting index model.** In the final model explaining the variability of beech and spruce

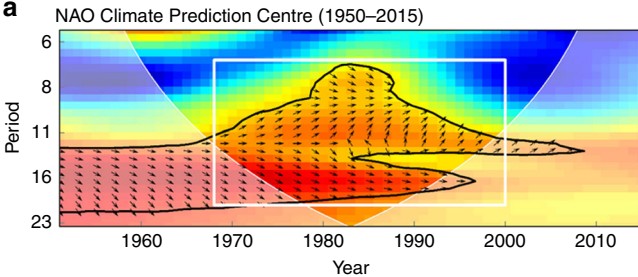

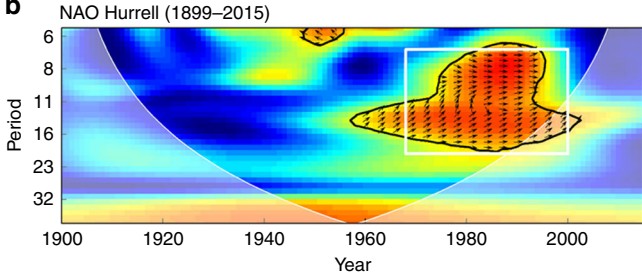

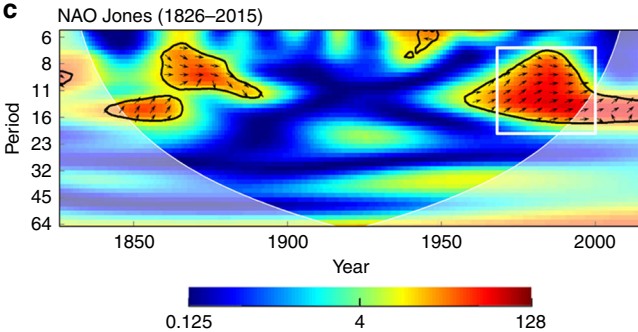

**Fig. 2** Wavelet coherence between the standardized beech M_index and winter-NAO indices. Wavelet coherence between the standardized beech M_index and winter-NAO indices. Winter-NAO indices used: Climate Prediction Centre-NOAA **a**, Hurrell[10] **b** and Jones et al.[64] **c**. X-axes: years of analysis. Y-axes: frequency domain of the NAO-masting relationship in years. Note that the x- and y-axes vary between plots. Arrows pointing up-right show in-phase behavior and y leading x, i.e., NAO leading M_index. Black contour designates frequencies of significant coherence ($p < 0.1$, two-sided test); the white cone of influence shows the data space immune from distortion by edge effects. The white squares show the period of strong coherence between 1960 and 2000

M_index, the high frequency (i.e., inter-annual) components of all seasonal NAO indices were significant ($p < 0.05$, two-sided test), except winter-NAO $Y_{M-1}$ for both species and the summer-NAO $Y_{M-2}$ for spruce (Table 2). Conversely, among the low-frequency (i.e., decadal trends) components, only winter-NAO was significant ($p < 0.001$) and displayed a strong positive effect on both species. Of all two-way interactions, low-frequency winter-NAO × high-frequency summer-NAO $Y_{M-1}$ was significant ($p < 0.05$, $\beta = +0.24$) in the beech model, but reduced the model Aikaike's information criterion (AIC) by only three points (Table 1). The beech model accurately described (pseudo-$R^2 = 0.59$) the observed M_index between 1952 and 2015 (Fig. 2), correctly reproducing most individual peaks (e.g., 1960, 1995, 2000, and 2014), and prolonged periods of high (e.g., 1989–1995) or low M_index (e.g., 1961–1985). The spruce model performed less well (pseudo-$R^2 = 0.42$) and failed to reproduce some peaks (e.g., 1974, 1995, 2004, 2006, and 2009). Models residuals generally showed no systematic bias and patterns (Supplementary Fig. 7), except that the precision of the model increased for higher AR1. This is expected, as after high masting there is often low

masting, but the opposite is not the case. The leave-one-out cross validation (LOOCV) was successful for beech ($r = 0.76$) but less successful for spruce ($r = 0.48$). However, both models accurately captured the shift in the frequency of large-scale masting events that occurred around 1985, from a period characterized by low M_index with relatively infrequent peaks, to a period of more regular large-scale masting events with high M_index. However, residual patterns (Fig. 1 and Supplementary Fig. 7) and LOOCV (Fig. 3) revealed that the model failed to predict the masting peak in 1958 for beech, highly underestimated the peaks in 1974, 2004, and 2011 for both species, and highly overestimated the low value in 1997.

**NAO–masting relationships and weather patterns**. The correlation between seasonal NAO indices and temperature and precipitation anomalies in Central-Northern Europe (Fig. 4) were consistent with weather patterns described in previous studies[10,12]. Positive winter-NAO was correlated to positive anomalies in both temperature and precipitation. Positive spring-NAO corresponded to mild temperatures and dry weather, whereas positive summer-NAO to positive anomalies in temperature and negative anomalies in precipitation throughout Central-Northern Europe.

## Discussion

In this study, we provide the evidence of a long-term relationship between masting in trees and inter-annual variation and decadal trends of a climate teleconnection. Several seasonal indices of NAO are jointly responsible for synchronizing beech and spruce masting in Central-Northern Europe. Although previous studies[7,15,17] focused on specific seasonal NAO series and regions (Table 1), our results show that NAO acts over multiple seasons and years synchronizing beech and spruce masting over a large part of their distribution, extending from 44°N–3.5°W to 58°N–26°E and 46°N–3.5°W to 61°N–30°E for beech and spruce, respectively. As a further step in comparison with previous studies, we tested the contribution of both the high- (inter-annual) and low- (decadal) frequency components of seasonal NAO indices. The group of significant high-frequency seasonal NAO predictors and the direction of their influence were consistent with previous findings based on raw seasonal NAO series. Negative NAO in the summer 2 years before fruit ripening, followed by a positive summer-NAO in the subsequent year[7] and by a positive spring-NAO during flowering[15], promotes beech masting. Spruce masting seems to be driven by the same high-frequency NAO components as for beech, except for a lack of influence of the summer two years before fruit ripening. Masting intensity of the previous year had a negative effect on both beech and spruce masting, which confirms that masting series display negative autocorrelation—indicative of resource depletion after large fruit crops[4,5]—even at a sub-continental scale, such as in the widespread masting failure in 1996, 1 year after the large masting event in 1995 (highlighted in Supplementary Fig. 1). Our models failed to predict poor masting in 1997; such overprediction decreased when we included an autoregressive term with a lag of −2 years in both models (−29 and −28% for beech and spruce, respectively). This term had a negative significant effect ($p < 0.05$, two-sided test) in both models, suggesting a long-lasting resource depletion after the large mast of 1995.

Among low-frequency components, only winter-NAO was significant, but displayed a strong positive effect on both beech and spruce masting in the frequency domain from 7 to 16 years. This shows that during prolonged periods of positive winter NAO, the occurrence of widespread masting events on beech and spruce increases. Although the model for beech performed better

**Table 2 Final regression model**

| Species | European beech | | | | Norway spruce | | | |
|---|---|---|---|---|---|---|---|---|
| Predictor | $\beta$ | SE | *p*-value | $\Delta$AIC | $\beta$ | SE | *p*-value | $\Delta$AIC |
| Autoregressive term | | | | | | | | |
| AR1 | −0.633 | 0.126 | 0.0001 | −20.11 | −0.266 | 0.129 | 0.0402 | −1.72 |
| High-frequency NAO | | | | | | | | |
| high summer-NAO $Y_{M-2}$ | −0.500 | 0.115 | 0.0001 | −15.37 | 0.158 | 0.116 | 0.17 ns | 0.14 |
| high winter-NAO $Y_{M-1}$ | 0.174 | 0.104 | 0.10 ns | −0.73 | 0.152 | 0.110 | 0.17 ns | 0.03 |
| high summer-NAO $Y_{M-1}$ | 0.373 | 0.109 | 0.0006 | −8.92 | 0.286 | 0.121 | 0.0180 | −3.54 |
| high spring-NAO $Y_M$ | 0.514 | 0.117 | 0.0001 | −16.33 | 0.364 | 0.122 | 0.0029 | −6.93 |
| Low-frequency NAO | | | | | | | | |
| low winter-NAO | 0.402 | 0.114 | 0.0004 | −10.00 | 0.407 | 0.121 | 0.0008 | −8.93 |
| low summer-NAO | −0.126 | 0.111 | 0.26 ns | +1.04 | −16.81 | 0.123 | 0.17 ns | 0.41 |
| low spring-NAO | −0.006 | 0.109 | 0.96 ns | +2.00 | −0.128 | 0.111 | 0.24 ns | 0.63 |
| Interaction | | | | | | | | |
| low winter-NAO xhigh summer-NAO $Y_{M-1}$ | 0.237 | 0.099 | 0.0170 | −3.26 | — | — | — | — |

Summary of the final regression model predicting the inter-annual variability of M_index of beech (period 1952–2015) and spruce (period 1959–2014) using both high- and low-frequency NAO components. Standardized coefficients are shown as model estimates ($\beta$) ± SE. $\Delta$AIC indicates the importance of the predictors and is calculated as the difference of AIC between the full model and the model without the predictor of interest. $Y_{M-2}$ and $Y_{M-1}$ indicate 2 and 1 years before fruit ripening, respectively, whereas $Y_M$ the masting year. ns = nonsignificant predictors.

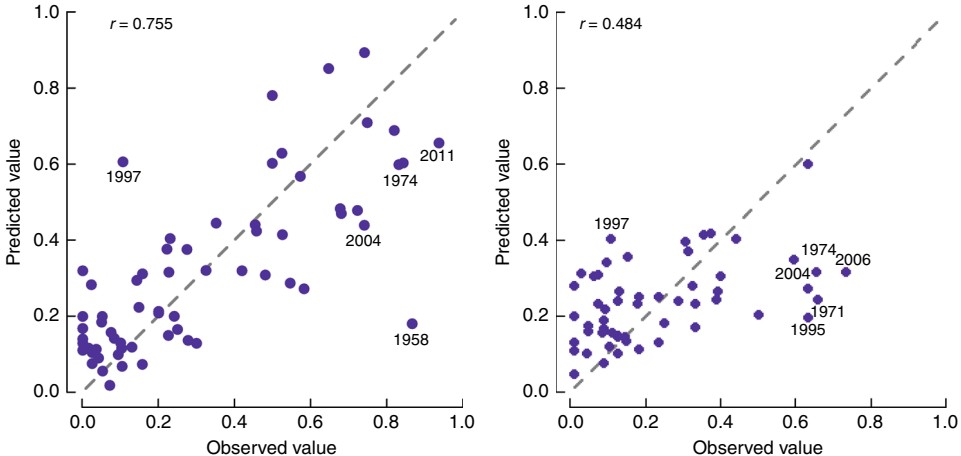

**Fig. 3** Leave one out cross-validation. Observed and predicted values of beech (left) and spruce (right) M_index from the LOOCV of the final model. The dashed line represents the perfect match between observed and predicted values. Years with the largest disparity are labeled individually

in comparison with the model for spruce, it is notable that disentangling the high-frequency (i.e., inter-annual) and the low-frequency (i.e., decadal) NAO components markedly improved the spruce model when compared to testing the raw NAO series (pseudo-$R^2$ of 0.42 vs. 0.27).

Many studies have discussed both the seasonal effects of NAO on Central-Northern European weather[10–12] and the effect of seasonal weather patterns on proximate causes of masting in the *Fagaceae* and *Pinaceae* families[3,7,19–22]. Our results highlight the link between seasonal NAO and weather patterns known to determine seed masting. Negative summer-NAO is associated with cool-wet summers in Central-Northern Europe (Fig. 4, see also Folland et al.[11] and Bladé et al.[12]), a weather pattern strongly correlated with beech masting when occurring 2 years before fruiting[3], and commonly interpreted as increasing available resources by enhancing litter mass loss and nutrient uptake due to high soil moisture[20,23–25]. In contrast, positive summer-NAO is associated with warm-dry summers in Central-Northern Europe (Fig. 4). This weather pattern is also correlated with both beech[3] and spruce[19] masting when occurring the year before fruit ripening, as it induces hormonal translocation for flower

primordial differentiation[21]. Finally, positive spring-NAO is associated with mild-dry weather (Fig. 4), which favours wind pollination and the related fruit-set in the seed production year[15,19,22]. With regards to the low-frequency component of winter-NAO, prolonged positive winter-NAO phases are associated with warm-wet winters (Fig. 4) with delayed positive effects on growing season temperatures[26]. Positive NAO in winter causes an earlier leafing out of beech in Central-Northern Europe[27], which lengthens the growing season. Moreover, positive winter-NAO enhances the primary production of Central-Northern European forests[28,29], which is indicative of available resources for reproduction in temperate trees[30]. Consequently, we speculate that during prolonged positive phases of the winter-NAO, such as in the early '50s and in '90s of the twentieth century[10], more resources were consistently available for beech and spruce masting throughout Central-Northern Europe.

Finally, the positive and significant interaction between the high-frequency component of summer-NAO of the year before masting and the low-frequency component of winter-NAO we found in the beech model could be interpreted as a higher sensitivity of the species to high temperatures inducing

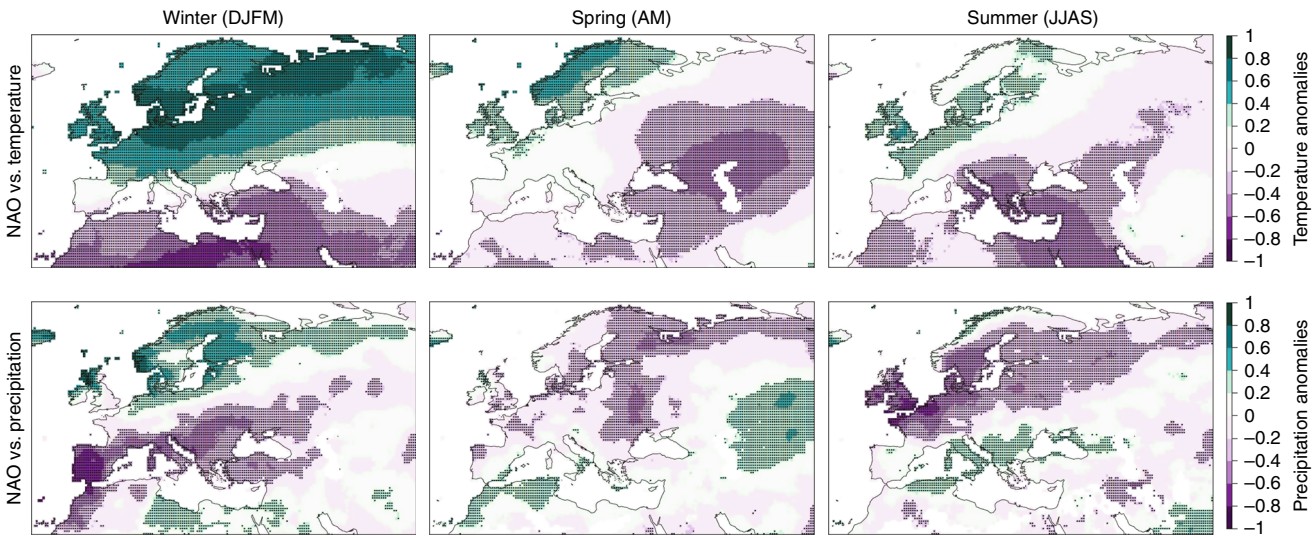

**Fig. 4** NAO and related weather patterns in temperature and precipitation. Correlation between NAO and temperature anomalies (first row), and between NAO and precipitation anomalies (second row) for the seasons winter, spring, summer (columns from left to right). Regions with significant correlations are denoted by black dots. Monthly precipitation and temperature have been obtained from the CRU database (version TS4.00). We aggregated these time series into seasonal time-series and the NAO indices according to our experiment design: winter (December–January–February–March, DJFM), spring (April–May, AM), and summer (June–July–August–September, JJAS). The period between 1950 and 2015 was considered for the correlation analysis. Figure created using ggplot2 package for R[67].

flowering during periods of increased resources[5,9]. However, the low ΔAIC of the interaction factor in the regression model, and the fact the interaction was not significant in the spruce model, advocates caution in interpreting this result. Indeed, few previous studies found such interaction, either in *Fagaceae*[20] or in other taxonomic groups[31]. In addition, in our study this interaction could be explained by a lagged effect of the winter-NAO on summer temperatures. Previous studies report that summer heat waves in Central-Northern Europe are strong and wide-ranging when positive summer-NAO occurs in years of positive winter-NAO[26]. For example, this was the case in 1994 before the beech masting in 1995, the largest event in the period 1952–2015 (Fig. 1). Interestingly, model residuals for year 1995 were reduced by including the interaction factor among predictors.

Whichever way this interaction is interpreted, our results show that seasonal and annual variations, and decadal trends in the NAO affect both short- and long-term patterns of tree masting in Central-Northern Europe, although these relationships are weak in some years and periods (i.e., non-stationary through time). Indeed, the NAO is the leading climate mode in Europe, but the Euro-North Atlantic region is also influenced by other large-scale atmospheric modes of variability, which oscillate at both inter-annual and decadal time scales replacing NAO influence on European weather patterns. This could explain model errors in given years, i.e., when the NAO-based models of both species had a lower explanative power (e.g., 1958, 2004). Here, weather patterns inducing masting might have been influenced by other broad-scale climate modes, such as the Scandinavian Pattern or the East-Atlantic and West Russian pattern[32], particularly in summer. For example, the summer 1957 was characterized by a severe heat wave in Central-Northern Europe, with locally record-breaking temperatures[33]. Although, generally, positive summer-NAO values are associated with high temperature in this region[12] (Fig. 4), the 1957 temperature-positive anomaly coincided with a below-average summer-NAO. The atmospheric patterns associated with the 1957-heatwave are instead attributed to two clusters of geopotential anomaly[34], a first extending over most of the

Scandinavian Peninsula and the second centred mostly over France and linked to the European summer blocking[35]. Similarly, the heatwave in summer 2003 coincided with a low summer-NAO index, and was attributed to tropical Atlantic forcing[35] in conjunction with a marked soil water deficit throughout the European continent[36].

From 1850 to 1900 and from 1960 to 2000, decadal winter-NAO and beech masting showed a significant coherence within a frequency domain of 7–16 years (Fig. 2), and a similar pattern was observed for the spring-NAO after 1985 (Supplementary Fig. 6 right). However, even the low-frequency components of winter- and spring-NAO were not always coherent with beech masting in the past two centuries. This is consistent with previous findings of a non-stationary influence of NAO over European weather patterns[10,35] and related ecological processes[13]. Our analyses confirm that in the long-term beech masting alternates between periods of frequent large-scale events and periods when such events are rare, thus generalizing the results of a previous study in Sweden[7]. We suggest this could be partly due to variability in the strength of NAO influence on the synchronization of weather patterns determining masting at the continental scale, which themselves appear to be largely stable through time[3].

Although we showed a non-stationary influence of NAO on tree masting at both inter-annual and decadal time scales, we also highlighted that NAO components have synchronized masting across Central-Northern Europe for long periods. According to our interpretation, NAO synchronizes the proximate causes of masting over large areas (Moran effect) at multiple stages of the reproductive cycle, such as resource accumulation, flower differentiation, and cross-pollination. This raises the question of the ecological meaning of the link between synchronous seed production and NAO patterns: is large-scale masting just a coincidental consequence of NAO controlling proximate causes, or does such synchronization also provide competitive advantages due to one or more economies of scale, hinting at the evolutionary factors of masting? Recently, Pearse et al.[5] theorised a complementarity between proximate (i.e., mechanism driving masting) and ultimate causes (i.e., evolutionary drivers) of masting. Several evolutionary hypotheses have been proposed as

ultimate causes of masting in beech and spruce, including the following: (i) pollination efficiency: percent of seed set is higher in high-flowering years[15,37]; (ii) predator-dispersal: seed predators are attracted to a large fruit crop increasing dispersal-related fitness benefits[38,39], particularly by birds[40,41]; and (iii) environmental prediction: masting anticipates favourable conditions for seedling establishment[19,42]. Notably, seasonal NAO patterns identified by this and other studies[7,15,17] are consistent with all these ultimate causes. Influences of positive spring-NAO on the pollination efficiency were already discussed by Fernández-Martínez et al.[15]. Regarding the predator-dispersal hypothesis, some studies found a positive relationship between winter-NAO and peaks in population dynamics of beech nuts dispersers such as *Columba palumbus*, *Fringilla coelebs*, *Pica pica*, and *Parus major*[43–45], and spruce dispersers such as *Carduelis spinus*[46]. Interestingly, a study in North America found that antiphased climate anomalies (i.e., dipoles such as NAO) modulate consistently both broad-scale seed-eating bird irruptions and widespread masting[14] resulting in birds anticipating the resource pulse, and this might be relevant also for beech and spruce seed dispersers.

A coherent picture can also be set for the environmental prediction hypothesis, although this theory has received less support, particularly when dealing with the prediction of future climate conditions favouring seedlings (e.g. wet seasons)[47]. Many studies demonstrated beech and spruce recruitment failure after a mast year because of thick litter or a closed tree canopy, typical of undisturbed stands[48,49]. However, regeneration is highly favored in both species by mixed severity disturbances[49–51], particularly when masting closely follows the disturbance[52–54]. Notably, periods of positive winter-NAO are associated with major damaging storms in Central-Northern Europe[55], whereas pressure patterns indicative of positive summer-NAO favor drought[10,26,35] and have been associated with fire activity in Southern Sweden[56] and Southern England[57]. Consequently, the teleconnection patterns that we show in this study to favour masting may also be associated with disturbance events that create conditions favoring seedling establishment. This is a different interpretation of the environmental prediction hypothesis, which does not require the prediction of future weather conditions criticised by previous studies[47].

After Pearse et al.[5], we suggest a coherent ecological-evolutionary theory using teleconnections as a bridge linking proximate and ultimate causes of masting. Previous studies used teleconnection indices to interpret the adaptive functioning of some ecological processes linked to climate oscillations[13] and this was proposed also for masting[16]. We do not imply that NAO has been the sole driver exerting a selective pressure on tree masting in Central-Northern Europe, as we found a non-stationary link between NAO and masting. Moreover, masting in the *Fagaceae* and *Pinaceae* families probably evolved before European weather and NAO relationships established. However, alternating large-scale features of atmospheric circulation with a significant impact on ecological and geophysical processes (e.g., resource cycles, animal populations, and natural disturbances) at any place and time might have exerted a selective pressure by influencing both the proximate and ultimate causes of masting.

Finally, our findings can improve the ability to predict masting based on NAO forecasts[58], particularly for beech, assuming that the non-stationarity in the NAO-masting relationships can be taken in consideration. To this extent, we suggest that studies linking climate change to tree masting should focus on how climate change affects NAO patterns[59] and their relationships with proximate causes of masting.

## Methods

**Masting index.** To analyse the response of beech and spruce masting at the sub-continental scale we computed an annual masting index for both species representing how much of the species distribution range displays a heavy seed crop in each year. We used the MASTREE database[18], which contains the longest available masting record for European beech and Norway spruce covering most of both species distribution and including annually resolved observations of beech and spruce seed production or related proxies (e.g., flowering, airborne pollen, fruiting, and dendrochronological reconstruction). We truncated the data to 2015 and 2014, and excluded the pollen and flowering proxies for beech and spruce respectively. We limited our analysis to the Central-Northern European regions, i.e., the core area of beech and spruce distribution where climate is most influenced by NAO[10–12], and where all previous studies were focused (Table 1). In total, we selected 5774 yearly observations from 337 series in 40 NUTS-1 (Nomenclature of Territorial Units for Statistics) for beech, and 5119 yearly observations from 191 series in 37 NUTS-1 for spruce (Supplementary Fig. 1). For each series, we extracted the five class ordinal masting index (from 1: very poor, to 5: very abundant) provided by MASTREE (column ORDmast). To avoid oversampling in regions with multiple data-series, we aggregated individual masting series into NUTS-1 chronologies by using the modal masting class for each year and NUTS-1. This produced long masting series with a minimum amount of missing data[3]. We then computed the annual masting index (M_index) for both species as the difference between the proportion of NUTS-1 displaying a good masting (classes 4 and 5, NC-45) and a poor masting (classes 1 and 2, NC-12) in any given year. M_index varies from − 1, when all NUTS-1 are in class 1 and 2 (broad masting failure), to + 1 when all NUTS-1 are in masting classes 4 and 5 synchronously. Intermediate (zero) values indicate a prevalence of class-3 masting, or a balanced distribution of NC-45 and NC-12 (Supplementary Fig. 1). M_index was computed on a different number of NUTS-1 each year, but covered a continuous period from 1760 to 2015 and 1800 to 2014 for beech and spruce, respectively (Supplementary Fig. 2).

**Raw seasonal NAO indices vs. masting index model.** To test at the sub-continental scale the consistency of NAO-mast correlations reported by previous studies at the regional scale (Table 1), we built a regression model. The building of this initial model was limited to 1952–2015 and 1959–2014 for beech and spruce respectively, when NUTS-1 chronologies were numerous (Supplementary Fig. 2), and evenly spread across the study area[3]. To build the predictors, we used the monthly NAO series provided by the Climate Prediction Center of the NOAA, which covers the entire period of analysis (i.e., 1950–2015). As in previous studies (Table 1), we computed raw seasonal NAO indices (i.e., without extracting high- and low-frequency components) by averaging monthly values as follows: December of the previous year to March (winter-NAO), April to May (spring-NAO), and June to September (summer-NAO). The shorter window for spring-NAO was justified by the fact that beech and spruce flowering and pollination in Central-Northern Europe occur mostly between April and May. We then fitted M_index of both species as a function of summer-NAO $Y_{M-2}$, winter-NAO $Y_{M-1}$ and summer-NAO $Y_{M-1}$, and spring-NAO $Y_M$. We accounted for the effect of resource depletion by previous masting events[4,5] by adding an autoregressive term with a lag of − 1 year. All predictors were standardized and checked for the absence of collinearity (pairwise Pearson correlation < 0.4). As the response was $\beta$-distributed for both beech and spruce (Supplementary Fig. 3, left column), we rescaled M_index from 0 to 1 and fitted a $\beta$-regression model via maximum likelihood using the R *betareg* package[60] for the R statistical framework.

**Low-frequency domain of NAO and tree masting relationships.** To test whether low-frequency changes in NAO influence beech and spruce masting, and to extract the frequency domain of such relationship, we carried out a wavelet coherence analysis using the *wtc* function in the R package *biwavelet*[61]. Wavelet analysis has often been applied to test for causality between teleconnections and geophysical processes[62], and allows assessment of whether such relationships are time-stationary across the frequency domain[63]. To avoid bias due to non-normality[63], M_index of both species was arcsine-transformed[64] before the wavelet analysis (Supplementary Fig. 3, right column). Wavelet analysis for spruce was limited to the period 1950–2014, as before 1950 there were too few data to calculate M_index (Supplementary Fig. 2 right), whereas for beech we used data extending back to 1826 (Supplementary Fig. 2 left). From 1950 to 2015 we used seasonal NAO indices by NOAA aggregating months as for the initial regression model. However, to extend the beech analysis before 1950 we used seasonal NAO indices provided by Hurrell[10] and Jones et al.[64], which cover the periods 1899–2015 and 1826–2015, respectively. The NOAA and Hurrell series are suited to test for coherence using all seasonal indices, because they are based on principal component analysis of sea level pressure over the North Atlantic[65]. Conversely, the Jones index is station-based (with fixed stations located in the Azores and Iceland), which makes it robust for winter NAO only[12]. Consequently, the wavelet coherence analysis for 1826–1899 was carried out on winter-NAO calculated from Jones et al.[64] only.

Each wavelet coherence analysis was computed using Morlet continuous wavelet transform and considering the lag − 1 autocorrelation of each series[63]. The data were padded with zeros at each end to reduce wraparound effects. Significance

of coherence within all frequency domains larger than five years (i.e., low frequencies) was tested using a time-average test and 500 Monte Carlo randomizations.

**High- and low-frequency NAO vs. masting index model**. To discriminate between low- and high-frequencies NAO components for the period 1952–2015, we fitted each of the three raw seasonal NAO series (i.e., winter-, summer-, and spring-NAO) with a running line smoother using the *supsmu* function of the *stats* R package. The span of all smoothers was set to the mean frequency domain at which wavelet coherence of raw seasonal NAO series against beech and spruce masting was significant (considering all NAO data sources and seasons). The corresponding high-frequency components were calculated by subtracting the smoothed series from the raw seasonal NAO index (Supplementary Fig. 4). For the summer season, two high-frequency summer-NAO components were calculated (i.e., for both $Y_{M-2}$ and $Y_{M-1}$). We then fitted M_index as a function of the three low-frequency components (winter-, summer-, and spring-NAO), the four high-frequency components (summer-NAO $Y_{M-2}$, winter-NAO $Y_{M-1}$, summer-NAO $Y_{M-1}$, and spring-NAO $Y_M$), and an autoregressive term with a lag of − 1 year. All predictors were standardized and checked for collinearity (pairwise Pearson's correlation < 0.4). The period of analysis was limited to 1952–2015 and 1959–2014 for beech and spruce, respectively, the response was assumed to be $\beta$-distributed, and the model was fitted via maximum likelihood using the *betareg* package for R[60].

When testing for interactions between all high- and low-frequency NAO predictors, we needed to limit overfitting, due to the large number of possible bivariate interactions relative to the number of observations. Following Quinn and Keough[66], we computed the residuals of the "null" model (without interactions), and fitted them as a linear function of each possible two-way interaction among standardized predictors. Only interactions producing a significant ($p < 0.05$) fit against null model residuals were added to the final model. The pseudo-$R^2$ of the model was computed as the squared correlation between the linear predictor for the mean and the link-transformed response[60].

We assessed the importance of each (standardized) predictor in the final model by calculating the difference between the AIC of the models with and without the concerned predictor ($\Delta$AIC)—the higher $\Delta$AIC, the larger the importance of the predictor in the model. The final model was validated by LOOCV.

**NAO relationships with weather patterns determining masting**. To test if NAO–masting relationships were coherent with weather patterns known to determine masting in both beech and spruce[3,19], we analysed the correlation between significant NAO patterns, as in the final regression model, and local weather anomalies. At each grid point of Europe, we computed the Spearman correlation between seasonal NAO indices and both precipitation and temperature series. Monthly precipitation and temperature were obtained from the Climate Research Unit database (version TS4.00). CRU time series and the NAO series were aggregated to the periods DJFM (December–January–February–March), AM (April–May), and JJAS (June–July–August–September), and linearly detrended. The period between 1950 and 2015 was considered for the correlation analysis.

**Code availability**. The R code used for analyses is provided as Supplementary material. We used R 3.3.1 version.

**Data availability**. The beech and spruce seed data that support the findings of this study are published in Ascoli et al.[18], are available on Ecological Archives (doi:10.1002/ecy.1785) and are accessible via the following link: http://onlinelibrary.wiley.com/store/10.1002/ecy.1785/asset/supinfo/ecy1785-sup-0002-DataS1.zip?v=1&s=2491b8cc559d5ec909f96dfc5a91397b1d7e9683. NAO data from the Climate Prediction Centre are available at: http://www.cpc.ncep.noaa.gov/products/precip/CWlink/pna/norm.nao.monthly.b5001.current.ascii.table; NAO data from Hurrell at the link: https://climatedataguide.ucar.edu/sites/default/files/nao_pc_monthly.txt; NAO data from Jones at the link: https://crudata.uea.ac.uk/cru/data/nao/nao.dat; CRU database (version TS4.00) is available at http://badc.nerc.ac.uk/data/cru/.

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

## Author contributions

D.A., A.H.P., and G.V. designed the research, analyzed the data, and wrote the manuscript. M.T. contributed to research design and interpreted weather data. I.D., M.C., J.M., and R.M. contributed to research design and data interpretation.

## Additional information

**Competing interests:** The authors declare no competing financial interests.

