## [Peer review file · Nature Communications]

Reviewers' comments:

Reviewer #1 (Remarks to the Author):

General comments

I liked this study. It goes a step further relative to previous studies relating teleconnections and masting. The study is interesting because it takes into account both low frequency (multi-annual/decadal) and high frequency (annual) variabilities, and analyses very long time series of masting. It thus improves our understanding on how climate variation affects masting.

I have, however, some concerns on possible drawbacks that the authors could consider before the article can be published in Nature Communications:

The study is conducted on only one species, *Fagus sylvatica*. Haven't you tested other species?

Several parts of the text are difficult to understand because low and high frequency variabilities are mixed. Already in the abstract itself the authors write: "In the last six decades, a three-year sequence of negative summer NAO, positive summer NAO, and positive spring NAO synchronized beech masting, together with a positive influence of winter NAO in the low-frequency domain (periodicity of 7-16 years).", requires a lot of attention of the reader, to say the least. So the text, here, but also throughout the whole article, should be re-written more clearly for the wide audience of Nature Communications. Pay special attention to the Results section at this regard.

More detailed comments:

Page 6: "There was no systematic bias, with no under- or over-prediction trend with AR1. This pattern is expected, as after high masting (positive value of AR1) there is often low masting, but the opposite is not the case. The leave-one-out cross validation (LOOCV) was successful ($r=0.76$)."
--- I think you in fact mean: negative AR1

Page 8: "In years when NAO has a lower explanative power (e.g. 1958, 2004), masting might have been influenced by other broad-scale climate modes in the previous years, such as the Scandinavian Pattern³³ or the East-Atlantic and West Russian pattern³⁴, particularly in summer".
May be also the WeMOI?

Page 9: "This is consistent with previous findings of a non-stationary influence of NAO over European weather patterns^{9,26} and related ecological processes^{12,15}." What do you really mean here? That the effects of NAO on environment change with time? Please clarify....if this was the case, you would be claiming for an invalidation of all the reconstructions conducted with tree-ring analyses.

Page 10: "A coherent picture can also be set for the environmental prediction hypothesis". Be careful with this hypothesis and this paragraph because this hypothesis has not many evidences supporting it (it is a little bit like science fiction).

Reviewer #2 (Remarks to the Author):

This is a large-scale observational and modeling study that links the North Atlantic Oscillation to mast events in European beech. I found the topic to be both exciting and currently important given the lively debate about the role of climate change (and climate in general) in masting. The statistical techniques used here were novel for the field of plant biology and masting and will add to that field.

My strongest concern and suggestion for the manuscript is that the correlation between the NAO and seed crops is interesting, but presumably divorced from mechanism by the step of understanding the particular weather patterns that actually determine seed crops. This has been done by several previous studies to varying degrees, and I would like to see the connection laid out here more explicitly. What aspect of weather does the NAO drive that in turn affects seed crops? Or does NAO alone out-perform estimates of weather patterns in determining seed crops? This is currently discussed (low NAO = wet, cold; high NAO = dry, warm), but I would like to see those connections made explicitly.

Smaller comments: In discussing the eco-evolutionary role of NAO connections, I think there are really two likely ways in which synchronizing on an oscillation might be important. These are 1) other important organisms like dispersers are synchronized by the same pattern (this is my interpretation of the current text), and 2) oscillations (by definition) are fairly predictable in their frequency, so if there is an optimal reproductive strategy to reproduce roughly every 8 years or something like that without being strictly oscillating (and thus predictable to seed predators, etc), climate oscillations provide an ideal set of cues to achieve some balance between a central tendency in frequency without strong predictability. I think this discussion is interesting. It is noteworthy that there seems to be a fairly small set of weather conditions that correlate with reproduction patterns in plants (i.e. spring/summer temperature, delta temperature, summer water, etc...though I guess this might also reflect a lack of search for other types of correlates...) and I like the hypothesis that climate oscillation patterns in part determine that subset.

I would like more discussion of the non-stationarity of the NAO – masting relationship, as I do not fully understand this.

Reviewer #3 (Remarks to the Author):

Review of manuscript NCOMMS-17-11511

"Inter-annual and decadal changes in teleconnections drive continental-scale synchronization of beech reproduction"

This manuscript aims to better understand the impacts of climate variability, as described by North Atlantic Oscillation (NAO), on *Fagus sylvatica* L. (European beech) masting. For this purpose, the author used a recently published long-term masting database covering most of the European beech distribution. The manuscript is nicely written and the methods used to assess both high- and low-frequency components of NAO and how their interaction affect beech masting seem to be appropriate. However, I'm convinced that the current manuscript does not provide sufficiently novelty to deserve publication in a high impact journal such as Nature Communications. The topics related with the impact of climate variability on different components of natural ecosystems has been for long time ago a theme that interested many researchers from different inter-disciplinary areas. In particular, several works have been done in the last decades relating NAO and other teleconnections with vegetation dynamics using different types of datasets (observations, remote sensing and tree rings, among other factors). The present work relies on a long-term dataset with around two centuries of observations and related proxies (e.g. flowering and airborne pollen). The very specific aspect of masting and the utilization of just one type of data (observational) restrict the public range for this work, what in my opinion is not compatible with the scope of Nature Communications. To deserve publication in Nature Communications other datasets should also be

used, such as the outputs of biophysical models. Thus, I believe the paper should not be accepted as it is. I strongly suggest the authors to try the publication of the work in another ISI journal.

Bellow, I listed some comments that could be important in the case of such option:

1. Abstract: In the abstract should be mentioned the dataset of masting used and the period covered.
2. Fig S5: there is no white boxes in A and C panels
3. Logistic Regression: The method of final regression model should be detailed in the main text. Please also explain the method used to compute pseudo-R² of logistic regression.
4. Lines 140-141: provide the level of significance.
5. Kelly 1994 was not numbered

Response to referees	
Referee#1	
The study is conducted on only one species, Fagus sylvatica. Haven't you tested other species?	After discussing the issue with the editor, we decided to test a NAO-masting model also for Picea abies (Norway spruce) and to integrate such additional analysis in the paper. Extending the analysis to other species was a challenging task. Indeed, large spatial and temporal data to reconstruct masting are currently available for beech and spruce only (MASTREE database, Ascoli et al. 2017). We checked alternative large databases for other species, such as the ICP forests (http://icp-forests.net/) or Herrera et al. (1998), Koenig and Knops (2000), Kelly and Sork 110 (2002), Schauber et al. (2002) and Kelly et al. (2013). However, these have a reduced number of observations at single species level. For example, mean observation number is 179 per species in Koenig and Knops (2000; see Table 1), versus >5000 observation for both beech and spruce in the MASTREE database. In addition, these alternative databases did not have a sub-continental scale and rarely exceed the time span of few decades (range in ICP is 2-12 years; range in Herrera et al. 1998, Table A1: 4-33 years; range in Kelly and Sork 2002: 6-35 years). Consequently, alternative databases (other than MASTREE) do not allow testing for decadal effects of climate oscillations at the continental scale, and the stationarity through time of teleconnection-masting relationships in particular. When submitting the manuscript to Nature Communications, we initially decided not to include spruce because of some constraints that we believed could reduce the clarity of the paper. However, following discussion with the editor and the advice of the Referee, we have managed now to include spruce without compromising the structure and the main message of the manuscript. Below we briefly outline the challenges in doing this, and the ways in which including this additional analysis has strengthened the manuscript. The data for beech is more extensive than the data currently available for spruce. In particular, the minimum spatial and temporal extent of data required to compute the spruce masting index (i.e., > 5 NUTS-1 per year) is available only for 1959-2014. This prevents us from testing for long-term relationships for spruce as we did for beech (i.e. over nearly two centuries), which was one of the most important and interesting elements of our study. Consequently, for spruce we are unable to explore the stationarity of the NAO-mast relationship to the same extent as for beech. However, models results for the last sixty years are consistent among the two species, with increased strength of the low-frequency association between beech and spruce masting and winter-NAO in recent decades. In our beech model, we made use of existing masting-NAO studies for this species. These previous studies used shorter datasets and were local/regional in scale, and we argued that they only captured part of the overall process. However, we used them to inform our initial model-building, helping to avoid the issue of model overfitting. Such an approach is not possible with spruce, due to lack of previous studies linking masting and NAO (limited to Fernández-Martínez et al., 2016, see lines 85-86). However, we used known relationships between climate-mast relationships for spruce, and NAO-climate relationships for Europe to justify the same starting point for the spruce model. Due to the shorter length of the spruce record, the model is fitted for the period 1959-2014. The application of the model to spruce was valuable, as similar NAO patterns seem to influence masting in both species. In particular, the decadal variation in winter NAO was positively and highly correlated to spruce masting, confirming one of the main study results. Consequently, after an exchange of ideas with the editor, we decided to fully integrate and discuss the spruce results (changes to the text are highlighted in light grey as requested by the editor) in order to address the suggestion of Ref#1 and

increase the novelty and impact of the manuscript. The inclusion of a second species reinforces our conclusion that tree masting at the continental scale is influenced by high- and low-frequency components of seasonal NAO, and that these relationships are non-stationary through time.

We thanks Ref#1 and the editor for challenging us in extending our analyses. We are convinced this further strengthened our manuscript now.

References

- Ascoli, D., Maringer, J., Hacket-Pain, A., Conedera, M., Drobyshev, I., et al. Two centuries of masting data for European beech and Norway spruce across the European continent. *Ecology*, 98(5), 1473 (2017).
- Fernández-Martínez, M., Vicca, S., Janssens, I.A., Espelta, J.M., Peñuelas, J. The North Atlantic Oscillation synchronises fruit production in western European forests. *Ecography*, 39, 1-11 (2016).
- Herrera, C.M., Jordano, P., Guitián, Traveset, J.A. Annual variability in seed production by woody plants and the masting concept: reassessment of principles and relationship to pollination and seed dispersal. *The American Naturalist*, 152(4), 576-594 (1998).
- Kelly, D., Sork V.L. Mast seeding in perennial plants: why, how, where? *Annual Review of Ecology and Systematics*, 33, 427-447 (2002).
- Kelly, D., Geldenhuis, A., James, A., Penelope Holland, E., Plank, M.J., Brockie, R.E., Cowan, P.E., Harper, G.A., Lee, W.G., Maitland, M.J., Mark, A.F., Mills, J.A., Wilson, P.R., Byrom, A.E. Of mast and mean: differential-temperature cue makes mast seeding insensitive to climate change. *Ecology Letters*, 16(1), 90-98 (2013).
- Koenig, W.D., Knops, J.M. Patterns of annual seed production by northern hemisphere trees: a global perspective. *The American Naturalist* 155(1):59-69 (2000).
- Schauber, E.M., Kelly, D., Turchin, P., Simon, C., Lee, W.G., Allen, R. B., Payton, I. J., Wilson, P.R., Cowan, P.E., Brockie, R.E. Masting by eighteen New Zealand plant species: the role of temperature as a synchronizing cue. *Ecology*, 83(5), 1214-1225 (2002).

Several parts of the text are difficult to understand because low and high frequency variabilities are mixed. Already in the abstract itself the authors write: "In the last six decades, a three-year sequence of negative summer NAO, positive summer NAO, and positive spring NAO synchronized beech masting, together with a positive influence of winter NAO in the low-frequency domain (periodicity of 7-16 years)." So the text, ... throughout the whole article, should be re-written more clearly for the wide audience of Nature Communications. Pay special attention to the Results section at this regard.

We agree with the Referee that several part of the manuscript were not clear, particularly as regards the distinction of the effects on masting of high- and low-frequency NAO components. We fully revised the abstract (lines 37-48) and several paragraphs in both the results (see lines 140-145) discussion (lines 174-181) and methods sections (lines 365-366) making the text more clear. In particular, every time we refer to a particular NAO season as masting predictor in the results section, we specify if it is a high- or low-frequency component (see for example line 140-145). Moreover, in order to clarify their meaning in several part of the text we used the terms inter-annual variation and decadal trends as synonymous for high- and low-frequency, respectively (see for example line 38-39). Changes to the text are highlighted in light grey as requested by the editor.

Page 6: "There was no systematic bias, with no under- or over-prediction trend with AR1. This pattern is expected, as after high masting (positive value of AR1) there is often low masting, but the opposite is not the case. The leave-one-out cross validation (LOOCV) was successful ($r=0.76$)."

--- I think you in fact mean: negative AR1

We removed "(positive value of AR1)" as this was not needed for the comprehension of the sentence that we rephrased (see lines 149-151).

Page 8: "In years when NAO has a lower explanative power (e.g. 1958, 2004), masting might have been influenced by other broad-scale climate modes in the previous years, such as the Scandinavian Pattern³³ or the East-Atlantic and West Russian pattern³⁴, particularly in summer".

May be also the WeMOI?

At the time of exploratory analyses, we tested several teleconnections indexes. The Euro-Atlantic region is mainly dominated by four large-scale atmospheric modes of variability: North Atlantic Oscillation (NAO), Eastern Atlantic (EA), Scandinavian (SCAND) and Eastern Atlantic-Western Russia (EAWR) patterns. Among these, NAO is the main mode of atmospheric variability in the North Atlantic region and has a strong impact on European climate (Yiou and Nogaj 2004, Casado et al. 2009, Bladé et al. 2012, Casanueva et al. 2014, Dunstone et al. 2016, Ceglar et al. 2017). For the Mediterranean, other regional indices have been developed (see e.g. Goodess and Jones 2002), as the Western Mediterranean Oscillation index (WeMOi) for the western part of the Mediterranean Sea (Martin-Vide and Lopez-Bustins 2006). For instance, Martín et al. (2012) have shown that the WeMOi is significantly better than the NAO to explain the climate impact of fish production in the western Mediterranean basin. However, the local character of the WeMOi index, suggests that it does not provide a better representation of the climate conditions affecting beech and spruce masting patterns in Central-Northern Europe, which is our main domain of study.

References

- Bladé, I., Liebmann, B., Fortuny, D., van Oldenborgh, G.J. Observed and simulated impacts of the summer NAO in Europe: implications for projected drying in the Mediterranean region. *Climate Dyn.*, 39, 709-727 (2012).
- Casado, M., Pastor, M., Doblas-Reyes, F. Euro-Atlantic circulation types and modes of variability in winter. *Theor. Appl. Climatol.*, 96, 17-29, (2009).
- Casanueva, A., Rodríguez-Puebla, C., Frías, M.D., González-Reviriego, N. Variability of extreme precipitation over Europe and its relationships with teleconnection patterns. *Hydrol. Earth Syst. Sci.*, 18, 709-725 (2014).
- Ceglar, A., Turco, M., Toreti, A., Doblas-Reyes, F.J. Linking crop yield anomalies to large-scale atmospheric circulation in Europe. *Agricultural and Forest Meteorology*, 240, 35-45 (2017).
- Dunstone, N., Smith, D., Scaife, A., Hermanson, L., Eade, R., Robinson, N., Andrews, M., Knight, J. Skilful predictions of the winter North Atlantic Oscillation one year ahead. *Nat. Geosci.*, 9, 809-814 (2016).
- Goodess, C.M., Jones, P.D. Links between circulation and changes in the characteristics of Iberian rainfall. *Int. J. Climatol.*, 22: 1593-1615 (2002).
- Martin-Vide, J., Lopez-Bustins, J.A. The western Mediterranean oscillation and rainfall in the Iberian peninsula. *Int. J. Climatol.*, 26, 1455-1475 (2006).
- Martín, P., Sabatés, A., Lloret, J., Martin-Vide, J. Climate modulation of fish populations: the role of the Western Mediterranean Oscillation (WeMO) in sardine (*Sardina pilchardus*) and anchovy (*Engraulis encrasicolus*) production in the north-western Mediterranean. *Climatic Change*, 110(3), 925-939 (2012).
- Yiou, P., Nogaj, M. Extreme climatic events and weather regimes over the North Atlantic: when and where? *Geophys. Res. Lett.*, 31(7), (2004).

Page 9: "This is consistent with previous findings of a non-stationary influence of NAO over European weather patterns^{9,26} and related ecological processes^{12,15}." What do you really mean here? That the effects of NAO on environment change with time? Please clarify....if this was the case, you would be claiming for an invalidation of all the reconstructions conducted with tree-ring analyses.

Yes, the effects of NAO on European weather patterns changes with time. The North Atlantic Oscillation (NAO) is the leading mode of atmospheric circulation in the North Atlantic region. However, NAO influence on European temperature and precipitation changes through time (i.e. it is non-stationary) at both inter-annual and decadal time scales. Indeed, changes in relationships between NAO and weather patterns have been found for some specific years (Cassou et al. 2005) and for prolonged periods (Beranová and Huth 2008) as a consequence of the shifting behavior of the NAO pressure centers (Vicente-Serrano and López-Moreno 2008). This means that in some years, the NAO does not influence temperature patterns, and if this happens for consecutive years, prolonged periods with a weak correlation between NAO and temperature variation in Europe may occur. This non-stationary influence of NAO on weather is reflected in changes in its correlation with ecological processes regulated by temperature and precipitation such as masting, or tree ring growth. Indeed, several studies have found a non-stationary effect of NAO on tree-ring width (Piovesan and Schirone 2008, Camarero 2011, Marcias et al. 2004). However, we do not claim for an invalidation of NAO reconstruction, as this was not the focus of our research. Tree rings do not respond to changes in pressure, and therefore tree-ring-based reconstructions of NAO are based on tree-ring responses to weather conditions associated with different phases of NAO. Reconstruction studies therefore require careful sample selection and assessment of the stability (stationarity) of relationships. The papers we cite here do not attempt at reconstructing NAO. They explore the link between forest/tree growth and NAO in regions where the NAO-climate relationships is complex and potentially unstable. We improved the discussion of the non-stationary influence of NAO (see lines 223-231) in order to make the concept now clearer to the reader.

References

- Beranová, R., Huth, R. Time variations of the effects of circulation variability modes on European temperature and precipitation in winter. *International Journal of Climatology*, 28(2), 139-158, (2008).
- Camarero, J.J. Direct and Indirect Effects of the North Atlantic Oscillation on Tree Growth and Forest Decline in Northeastern Spain. In. *Hydrological, Socioeconomic and Ecological Impacts of the North Atlantic Oscillation in the Mediterranean Region* pp 129-152 (2011).
- Macias, M., Timonen, M., Kirchhefer, A. J., Lindholm, M., Eronen, M., Gutiérrez, E. Growth variability of Scots pine (*Pinus sylvestris*) along a west-east gradient across northern Fennoscandia: A dendroclimatic approach. *Arctic Antarctic And Alpine Research*, 36, 565-574 (2004).
- Cassou, C., Terray, L., Phillips, A.S. Tropical Atlantic influence on European heat waves. *Journal of Climatology*, 18(15), 2805-2811 (2005).
- Piovesan, G., Schirone, B. Winter North Atlantic oscillation effects on the tree rings of the Italian beech (*Fagus sylvatica* L.). *International Journal of Biometeorology*, 44(3), 121-127, (2000).
- Vicente-Serrano, S. M., López-Moreno, J.I. Nonstationary influence of the North Atlantic Oscillation on European precipitation. *Journal of Geophysical Research Atmospheres*, 113(D20), (2008).

Page 10: "A coherent picture can also be set for the environmental prediction hypothesis". Be careful with this hypothesis and this paragraph because this hypothesis has not many evidences supporting it (it is a little bit like science fiction).

We acknowledge that the environmental prediction hypothesis claiming that plants are able to predict future climate conditions favorable for seedling establishment has received little support so far. This was demonstrated by Koenig et al. (2010), that we now discuss and included in the reference list of the revised manuscript (see lines 274-276 and 285). However, the idea we suggest departs from the classical "environmental prediction hypothesis". We argue that the conditions favorable for seedling emergence and establishment anticipate or concur with masting. Indeed, wind storms driven by positive phases of the winter NAO occur during periods of higher seed production and are known to modify the forest structure, opening gaps that are needed for beech and spruce recruitment. However, our point is not to claim that the environmental prediction hypothesis is

	the evolutionary mechanism that selected masting in beech and spruce, as well as other hypotheses such as the pollination efficiency or the predator-dispersal. Our point is to highlight that NAO drives coherently both proximate causes (i.e. masting cues) and potential ultimate causes (i.e. evolutionary factors), as theorized by Pearse et al. (2017), and we believe this is an interesting idea that can inspire new research. We improved the related text in the manuscript in order to make this idea (see lines 286-295) now clearer to the reader. References Koenig, W.D., Knops, J.M., Carmen, W.J. Testing the environmental prediction hypothesis for mast-seeding in California oaks. Canad. J. of For. Res., 40(11), 2115-2122 (2010). Pearse, I.S., Koenig, W.D., Kelly, D. Mechanisms of mast seeding: resources, weather, cues, and selection. New Phytol., 212(3), 546-562 (2016).
Referee#2 My strongest concern and suggestion for the manuscript is that the correlation between the NAO and seed crops is interesting, but presumably divorced from mechanism by the step of understanding the particular weather patterns that actually determine seed crops. This has been done by several previous studies to varying degrees, and I would like to see the connection laid out here more explicitly. What aspect of weather does the NAO drive that in turn affects seed crops? This is currently discussed (low NAO = wet, cold; high NAO = dry, warm), but I would like to see those connections made explicitly.	We think that this is a very good point. Following this suggestion, we analyzed the correlation between seasonal NAO indexes and local weather patterns that are known to affect beech and spruce masting (Picea abies analyses were added after a request by the Referee#1). Namely, weather patterns are: high temperature and precipitation in winter; low temperature and abundant precipitation in summer in the previous two years (for beech only); high temperature and low precipitation in summer the year before masting, and lack of frost and dry weather in spring the year of masting. These are the main weather cues observed in previous studies and in Vacchiano et al. (2017), cited in the text, we recently published where we demonstrate that weather cues of masting are consistent through space and time. To address Referee#2 request, we performed a spatial analysis, computing the Spearman correlation between NAO and precipitation and temperature series at each grid point in Europe (see methods section at line 391-399). We therefore included a new main figure in the manuscript (Figure 5) showing that the correlation between seasonal NAO indexes and temperature and precipitation anomalies are consistent with weather patterns affecting seed masting in beech and spruce, as evidenced by previous studies. At line 95 we added a new study objective related to this extended analysis. We warmly thank the Referee for suggesting such additional analysis. The new results and the discussion on their implications to our results (see line 192-210) represent an added value to the manuscript. All changes to the text are highlighted in light grey as requested by the editor. References Vacchiano, G., Hackett-Pain, A., Turco, M., Motta, R., Maringer, J., et al. Spatial patterns and broad-scale weather cues of beech mast seeding in Europe. New Phytol., 215(2), 595-608 (2017).
In discussing the eco-evolutionary role of NAO connections, I think there are really two likely ways in which synchronizing on an oscillation might be important. These are 1) other important organisms like dispersers are synchronized by the same pattern (this is my interpretation of the	The hypothesis proposed by the Referee#2, that climate oscillation provide an ideal set of cues for reproduction in plants, deserves further discussion. Points to discuss are two: 1. Climate-oscillations determine the subset of masting cues, 2. The masting strategy benefits of a central tendency in frequency without strong predictability (to avoid predation), and climate oscillations are an ideal cue in this sense since they display a periodicity that varies through time. 1. As regards the first point, most studies linking climate-oscillations to masting point to a mechanism in which the teleconnection determines a warm-dry weather, which in turn triggers masting. The El Niño Southern Oscillation correlates to large-scale masting in several taxa in East Asia and Oceania. Most studies attribute this correlation to warm-dry weather driven by ENSO cycles one year to few months

current text), and 2) oscillations (by definition) are fairly predictable in their frequency, so if there is an optimal reproductive strategy to reproduce roughly every 8 years or something like that without being strictly oscillating (and thus predictable to seed predators, etc), climate oscillations provide an ideal set of cues to achieve some balance between a central tendency in frequency without strong predictability. I think this discussion is interesting. It is noteworthy that there seems to be a fairly small set of weather conditions that correlate with reproduction patterns in plants (i.e. spring/summer temperature, delta temperature, summer water, etc) and I like the hypothesis that climate oscillation patterns in part determine that subset.

before masting in both East Asia (Ashton et al. 1988, Sakai et al. 2006) and in New Zealand and Tasmania (Schauber et al. 2002, Fletcher 2015). Similarly, in North-America masting in conifers is favored by a warm, dry spring and summer driven by a positive phase of the North-Pacific index in the year before masting (Strong et al. 2015). In Europe we show that a warm-dry summer the year before masting due to positive NAO favors beech and spruce masting. Consequently, according to available studies linking large-scale climate to masting, the set of cues seems to be fairly small, indeed, and point to temperature as the main driver of masting. Notably, temperature is the most geographically consistent factor determined by large-scale climate modes, overriding effects of site conditions, given the absence of a strong altitudinal gradient. Solar radiation is also mentioned in some studies, but rarely tested. Notably, teleconnection phases conducive to warm temperatures and dry conditions are characterized by high pressure patterns over extent areas with a reduced cloudiness and a subsequent higher irradiance for prolonged periods.

2. As regards the second point, the literature on the periodicity of climate oscillations show a relatively narrow range of periods, i.e. ENSO periodicity ranges from 2-3 to 11-12 years; NAO periodicities range from 3-6 years to 8-16 (Olesen et al. 2012). Our results show that NAO and tree masting display a wavelet coherence ranging from 7 to 16 years which might reflect changes in the periodicity of NAO. This support the Referee#2 idea that masting fluctuation driven by NAO periods is not predictable, and that a central tendency can be set (in our study was 11 years).

References

Ashton, P.S., Givnish, T.J., Appanah, S. Staggered flowering in the Dipterocarpaceae: new insights into floral induction and the evolution of mast fruiting in the seasonal tropics. *Am. Nat.*, 132(1), 44-66 (1988).
Olsen, J., Anderson, N.J., Knudsen, M.F. Variability of the North Atlantic Oscillation over the past 5,200 years. *Nature Geoscience*, 5(11), 808 (2012).
Strong, C., Zuckerberg, B., Betancourt, J.L., Koenig, W.D. Climatic dipoles drive two principal modes of North American boreal bird irruption. *Proc. Natl. Acad. Sci. U.S.A.*, 112(21), E2795-E2802 (2015).
Schauber, E.M., Kelly, D., Turchin, P., Simon, C., Lee, W.G., Allen, R.B. Masting by eighteen New Zealand plant species: the role of temperature as a synchronizing cue. *Ecol.*, 83(5), 1214-1225 (2002).

I would like more discussion of the non-stationarity of the NAO – masting relationship, as I do not fully understand this.

Previous research showed that NAO influence on European temperature and precipitation changes trough time (i.e. it is non-stationary) at both inter-annual and decadal time scales (Cassou et al. 2005, Vicente-Serrano and López-Moreno 2008). In some years, the NAO does not influence temperature patterns, and if this happen for consecutive years, prolonged periods with a weak correlation between NAO and temperature variation in Europe may occur. This is a consequence of the shifting behavior of the NAO pressure centers (Beranová and Huth 2008, Vicente-Serrano and López-Moreno 2008). The non-stationary influence of NAO on weather has consequences on the correlation between NAO and the ecological processes determined by temperature and precipitation patterns, such as tree ring growth (e.g. Piovesan and Schirone 2000) or masting. We improved the discussion of the non-stationary influence of NAO (lines 223-231) and we hope now the concept is clearer.

References

Beranová, R., Huth, R. Time variations of the effects of circulation variability modes on European temperature and precipitation in winter. *International Journal of Climatology*, 28(2), 139-158 (2008).
Cassou, C., Terray, L., Phillips, A.S. Tropical Atlantic influence on European heat waves. *Journal of Climatology*, 18(15), 2805-2811 (2005).
Piovesan, G., Schirone, B. (2000). Winter North Atlantic oscillation effects on the tree rings of the Italian beech (*Fagus sylvatica* L.). *International Journal of*

	Biometeorology, 44(3), 121-127. Vicente-Serrano, S. M., & López-Moreno, J.I. Nonstationary influence of the North Atlantic Oscillation on European precipitation. Journal of Geophysical Research: Atmospheres, 113(D20) (2008).
Referee#3	
I'm convinced that the current manuscript does not provide sufficiently novelty to deserve publication in a high impact journal such as Nature Communications. The topics related with the impact of climate variability on different components of natural ecosystems has been for long time ago a theme that interested many researchers from different inter-disciplinary areas. In particular, several works have been done in the last decades relating NAO and other teleconnections with vegetation dynamics using different types of datasets (observations, remote sensing and tree rings, among other factors). The present work relies on a long-term dataset with around two centuries of observations and related proxies (e.g. flowering and airborne pollen). The very specific aspect of masting and the utilization of just one type of data (observational) restrict the public range for this work, what in my opinion is not compatible with the scope of Nature Communications. To deserve publication in Nature Communications other datasets should also be used, such as the outputs of biophysical models.	We extended our analyses by including a new tree species (Picea abies, spruce) and by making explicit the link between seasonal NAO indexes and weather patterns known to determine masting in both beech and spruce. The application of the model to spruce was valuable, as similar NAO patterns seem to influence masting in both species. In particular, the decadal variation in winter NAO was positively and highly correlated to spruce masting, confirming one of the main study results. In addition, we included a new figure in the manuscript (Figure 5) showing that the correlation between seasonal NAO indexes and temperature and precipitation anomalies in the last 60 years are consistent with weather cues affecting seed masting in beech and spruce, as evidenced by previous studies. We hope the Referee will appreciate our effort to increase the novelty of our study. In our opinion, this paper presents a major advancement in the understanding of large-scale climate influences on ecosystem processes. We provide the longest and the most spatially extensive analysis of the relationship between tree masting series and climate teleconnections, analyzing for the first time inter-annual patterns as well as decadal trends, and the stationarity of these relationships. Notably, disentangling high- and low-frequency NAO components has never been attempted in previous studies linking plant ecology to NAO (e.g. Post and Stenseth 1999, Ottersen et al. 2001, Mysterud et al. 2003). In addition, we provide a novel and holistic interpretation of large-scale masting and climate relationships, bridging multi-scale causes of masting and interpreting them in an evolutionary context. We believe this is a major step forward in tree seeding biology supported by a new multi-century-long dataset at large (sub-continental) scale using different type of data (seed, fruit, pollen), including dendrochronological reconstruction of masting (see Ascoli et al. 2017, and line 308). We focused on only two tree species because data to carry out such analysis (>5,000 individual observations for each species covering two centuries) are available for European beech and Norway spruce only. Our study defines and statistically tests a set of hypotheses on these two very large datasets and in this way guides future applications of physiological models to study masting. We are confident that the manuscript would be of wide interest to readers of Nature Communications since large-scale masting has major cascading effects on food webs, providing large quantities of pollen for insects and seeds for animals, and on human health, because of pollen allergies and epidemic diseases vectored by frugivorous mammals. In addition, understanding large-scale masting patterns helps us to interpret climate reconstruction by tree-ring analyses or animal population dynamics and migrations. Our findings improve the ability to predict masting and related cascading effects based on North Atlantic Oscillation forecasts. Studies linking climate change to seed production could now focus on how climate change affects teleconnection patterns and their relationships with masting so far. References Ascoli, D., Maringer, J., Hackett-Pain, A., Conedera, M., Drobyshev, I., et al. Two centuries of masting data for European beech and Norway spruce across the European continent. Ecology, 98(5), 1473 (2017). Mysterud, A., Stenseth, N. C., Yoccoz, N. G., Ottersen, G., Langvatn, R. The response of terrestrial ecosystems to climate variability associated with the North Atlantic Oscillation. The North Atlantic oscillation: climatic significance and environmental impact, 235-262, (2003). Ottersen, G., Planque, B., Belgrano, A., Post, E., Reid, P.C., Stenseth, N.C. Ecological effects of the North Atlantic oscillation. Oecologia, 128(1), 1-14, (2001).

	Post, E., Stenseth, N.C. Climatic variability, plant phenology, and northern ungulates. Ecology , 80(4), 1322-1339, (1999).
Fig S5: there is no white boxes in A and C panels.	We removed the reference to the white box for graphs A and C. Thanks.
Logistic Regression: The method of final regression model should be detailed in the main text. Please also explain the method used to compute pseudo-R2 of logistic regression.	In the betareg package for R, the pseudo R-squared is the one suggested by Cribari-Neto et al. (2010) for beta regressions: the squared correlation between the linear predictor for the mean and the link-transformed response. This explanation is now added to the text (see line 384-385). Reference Cribari-Neto, F., Zeileis, A. Beta Regression in R. J. Stat. Softw. , 34(2), 1-24 (2010).
Abstract: In the abstract should be mentioned the dataset of masting used and the period covered.	The database used for the analysis was added to the abstract (line 40). We were not able to also include the period covered for both European beech and Norway spruce due to the words limit (150) required by the journal.
Lines 140-141: provide the level of significance	The level of significance was provided (see line 141).
Kelly 1994 was not numbered	We removed Kelly 1994, as this reference was redundant and we needed to reduce the number of references. Indeed, the same definition of proximate and ultimate causes of masting are included in the reference Pearse et al. 2016 (line 259, Ref. number 5, already cited in the text and including Kelly D. as co-author), who recently addressed the relationship between proximate and ultimate causes of masting.

REVIEWERS' COMMENTS:

Reviewer #1 (Remarks to the Author):

The authors have made a thorough revision of the manuscript based on the three reviewers' comments.

Necessary changes have been done, including extending the analysis to other species and the spatial analysis.

Acceptable explanations have been done too for not changing some items.

The manuscript is now in my option acceptable for publication since it provides a valuable long and spatially extensive analysis of the relationship between tree masting and climate teleconnections, while considering inter-annual as well as decadal trends.

If the authors wish to improve it a little bit more, they could consider better clarifying the text in a final run, for example to better distinguish the effects of high- and low-frequency NAO components on masting, or the temporal changes in the effects of NAO on European weather patterns.

I would skip all expressions claiming "first time" "never before done".....They are very risky and usually not acceptable in scientific manuscripts. At this regard, Fernández-Martínez et al., 2016 in their recent paper already dealt with some of these issues.

Reviewer #2 (Remarks to the Author):

I enjoyed the revision of this manuscript. Strengthening the connection between NAO, weather patterns, and seed production has made the manuscript more interesting to me, and allows me to place this manuscript among the various work that has been done on European beech and spruce. The addition of the spruce analysis was also a good complement to the beech data. I do not have any major comments on the manuscript as it stands, but I will put in a plug for one research direction that is probably better suited for another forum.

As work progresses on this subject, I would love to see the mechanistic link between NAO – weather – seed set strengthened. As it currently stands the connection between NAO – seed set is well analyzed and reported here. The NAO-weather connection is analyzed here and more rigorously elsewhere, and citations are given for the relationship between weather and seed set. Ultimately, I would find a mechanistic model (path analysis or such) to be the most convincing unifying piece of evidence showing the influence of NAO on seed set via particular weather. I can see that this type of analysis is not suited for the current manuscript, as it would be long and rather complicated, particularly being that one single aspect of weather may not be the sole driver of seed set. But exploring this complex relationship would move the field beyond the observation of a mind-boggling connection between climatic oscillations and ecological events to a nuanced understanding of the chain of events that actually connect the two.

Minor comments:

L57: mechanisms

L81: such -> this

L86: needs more description of what is meant here. Perhaps "...sole observed correlation between NAO and seed set in spruce..."

L95: coherent -> consistent

Reviewer #3 (Remarks to the Author):

I have read the revised version of the paper with interest, and looked carefully to the additional analyses presented. The new dataset for masting of Norway spruce and the maps showing the control of NAO on temperature and precipitation have now become part of the main text and supplement. These new findings have resulted from a hard effort to increase the quality of this work, but they went beyond what I have suggested in my review. However, the new results presented with the new dataset are not very exciting, as the model do not show results with the same quality of the previous ones.

Much of the new information you have presented supports the view as presented in your paper, but I think that my initial concerns about the novelty of the present work has not been taken away 100%. I recognize the add value of the long-term dataset but this dataset was already published. I agree about the novel aspect of disentangling high- and low-frequency NAO components and about the holistic interpretation of large-scale masting and climate relationships that was achieved in the paper. However, in order to provide robust results you should compare your findings with the ones obtained from the application of your methodology to different type of data (seed, fruit, pollen), including dendro chronological reconstructions. I hope the authors will consider such an addition.

Moreover, the new results of Figure 5 do not seem novel enough as the driver effect of NAO on temperature and precipitation has been presented since the definition of NAO Index in tens or hundreds papers. The novelty could be related with the definition used for seasons but those do not seem to be novel enough to be published in Nature Communications.

Response to reviewers	
Referee#1	Response
If the authors wish to improve it a little bit more, they could consider better clarifying the text in a final run, for example to better distinguish the effects of high- and low-frequency NAO components on masting, or the temporal changes in the effects of NAO on European weather patterns.	We attempted to better distinguish the effects of high- and low-frequency NAO components on masting, or the temporal changes in the effects of NAO on European weather patterns throughout the manuscript, e.g. LN97-98, LN195-196.
I would skip all expressions claiming “first time” “never before done”.....They are very risky and usually not acceptable in scientific manuscripts	All expressions claiming the novelty of the study were removed. In particular, “first” was removed at LN 166. “novel” was removed at LN 172.
Referee#2	
As work progresses on this subject, I would love to see the mechanistic link between NAO – weather – seed set strengthened. As it currently stands the connection between NAO – seed set is well analyzed and reported here. The NAO-weather connection is analyzed here and more rigorously elsewhere, and citations are given for the relationship between weather and seed set. Ultimately, I would find a mechanistic model (path analysis or such) to be the most convincing unifying piece of evidence showing the influence of NAO on seed set via particular weather. I can see that this type of analysis is not suited for the current manuscript, as it would be long and rather complicated, particularly being that one single aspect of weather may not be the sole driver of seed set. But exploring this complex relationship would move the field beyond the observation of a mind-boggling connection between climatic oscillations and ecological events to a nuanced understanding of the chain of events that actually connect the two.	We agree with the reviewer that a mechanistic model using structural equation modeling (which includes path analysis) to examine the relationships between NAO variables, weather variables and reproductive effort and growth, would provide a better understanding of the chain of events that connects teleconnections to seed masting in plant species. We also agree that this analysis is not suited for the current manuscript, but we thanks the reviewer for this idea that we are willing to address in the next future with further research studies.
L57: mechanisms	Thanks. Done, now at LN 56.
L81: such -> this	Done. Now at LN 80.
L86: needs more description of what is meant here. Perhaps “...sole observed correlation between NAO and seed set in spruce...”	Changed according the reviewer suggestion. Now at LN 85.
L95: coherent -> consistent	“Coherent” was changed in “consistent”.

Referee#3	
The new results presented with the new dataset are not very exciting, as the model do not show results with the same quality of the previous ones.	We do not share the rather pessimistic view of the reviewer on the additional results involving spruce. We agree that these were less statistically strong. However, they do appear convincing on the absolute scale (pseudo-R2 for predicting model = 0.42). Importantly, spruce masting patterns were broadly consistent with the ones revealed in beech in every analysis involving both species. In particular, both models succeeded in capturing the shift in the frequency of large-scale masting events that occurred around 1985. The results of analyses involving spruce do thus support conclusions obtained on beech data strengthening the key message of our study.
In order to provide robust results you should compare your findings with the ones obtained from the application of your methodology to different type of data (seed, fruit, pollen), including dendrochronological reconstructions. I hope the authors will consider such an addition.	The reviewer rises a legitimate point here. We agree that verification of the patterns on a different dataset and across time frequency ranges could indeed provide additional line of support for the reported patterns. However, we strongly believe that the theoretical and analytical challenges associated with this exercise make it difficult to accommodate this request within the scope of the current manuscript. First, to the best of our knowledge, there are no datasets on seed counts nor on pollen abundances covering the time span and reaching the geographical coverage of the datasets (MASTREE) used in this study. The few datasets not included in MASTREE, e.g. the pollen data of the Réseau National de Surveillance Aérobiologique (RNSA) in France, combining data from about 90 stations of the continental France, or the ICP Forests inventory (http://icp-forests.net/) are just about one decade long, which precludes their use as a validation dataset, particularly for the low-frequency component. If the reviewer has in mind pollen data developed from sediment based reconstructions, the temporal resolution could be an issue here, specifically for the high-frequency component of our model: even with AMS dating the sediment data of non-laminated sediments are at 20-30 year resolution at best. Furthermore, all available dendrochronological reconstructions for the two species were included in MASTREE. New dendrochronological reconstructions could indeed serve as good validation tools, particularly suitable for testing low- and high-frequency effects of climate variability. However, at the moment this data type lacks adequate spatial coverage. We are actually quite excited by the reviewer remark, since we in fact currently collect long beech chronologies to attempt such reconstruction across European sub-continent. This follow-up study involves parametrization of logistical transfer function and a check of parametrization consistency across Europe (alternatively, development of region-specific parameterizations). This work, however, is a study of its own and clearly is outside

	the scope of the current manuscript.
Moreover, the new results of Figure 5 do not seem novel enough as the driver effect of NAO on temperature and precipitation has been presented since the definition of NAO Index in tens or hundreds papers. The novelty could be related with the definition used for seasons but those do not seem to be novel enough to be published in Nature Communications.	We agree with the reviewer that the message reported in Figure 5 is not new, except for the spring season. The figure help in understanding the key message of additional analyses requested by the reviewer#2 and we believe it helps the reader in understanding key links between NAO and weather patterns influencing tree masting synchronization at the large scale. We wish therefore to keep Figure 5 in the main text.